

# Tartrazine induces structural and functional aberrations and genotoxic effects *in vivo*

Latifa Khayyat[1], Amina Essawy[2], Jehan Sorour[2] and Ahmed Soffar[2]

[1] Biology Department, Faculty of Applied Sciences, Umm Al-Qura University, Makkah, Saudi Arabia
[2] Zoology Department, Faculty of Science, Alexandria University, Alexandria, Egypt

## ABSTRACT

Tartrazine is a synthetic organic azo dye widely used in food and pharmaceutical products. The current study aimed to evaluate the possible adverse effect of this coloring food additive on renal and hepatic structures and functions. Also, the genotoxic potential of tartrazine on white blood cells was investigated using comet assay. Twenty adult male Wistar rats were grouped into two groups of 10 each, control- and tartrazine-treated groups. The control group was administered orally with water alone. The experimental group was administered orally with tartrazine (7.5 mg/kg, b.wt.). Our results showed a marked increase in the levels of ALT, AST, ALP, urea, uric acid, creatinine, MDA and NO, and a decreased level of total antioxidants in the serum of rats dosed with tartrazine compared to controls. On the other hand, administration of tartrazine was associated with severe histopathological and cellular alterations of rat liver and kidney tissues and induced DNA damage in leucocytes as detected by comet assay. Taken together, the results showed that tartrazine intake may lead to adverse health effects.

## INTRODUCTION

Tartrazine (E number E102) a synthetic azo dye with lemon yellow color, is a commonly used food colorant for food products that we eat almost every day (*Mittal, Kurup & Mittal, 2007*). Among foods containing tatrazine are soft drinks and sport drinks, flavored chips, sauces, ice creams, jams, jellies and chewing gums (*Walton et al., 1999*). Tartrazine is found in many non-food consumables such as soaps, cosmetics, shampoos, vitamins and certain prescription medications (*Amin, Abdel Hameid & AbdElsttar, 2010*). Moreover, it is used in many developing countries as a low cost alternative for saffron in cooking (*Mehedi et al., 2009*).

Tartrazine toxicity results directly or indirectly from the metabolic reductive biotransformation of the azo linkage (*Chequer, Dorta & De Oliveira, 2011*). For example, tartrazine can undergo metabolic reduction in the intestine of the animal by the intestinal microflora, thus resulting in formation of two metabolites, sulfanilic acid and aminopyrazolone (*Chung, Stevens & Cerniglia, 1992*). These metabolites of tartrazine can

Corresponding author
Amina Essawy,
amina_essawy@yahoo.com

generate reactive oxygen species (ROS), generating oxidative stress, and affect hepatic and renal architectures and biochemical profiles (*Himri et al., 2011*).

Several *in vivo* studies have been administered tartrazine in different doses and revealed neither cytotoxic changes in tissues and organs nor development of neoplastic alteration in experimental animals (*Rus et al., 2010*).

Since literature data regarding the toxicity of tartrazine are contradictory, the present work aims to evaluate the toxic potential of tartrazine on livers and kidneys of male albino rats. In addition, this study was extended to evaluate the effect of this food colorant on the biomarkers of oxidative stress and to examine its genotoxic effect on white blood cells using comet assay.

## MATERIALS AND METHODS

Twenty adult male Wistar albino rats of 146–153 g body weight were used in this study. They were kept under observation for about one week before the beginning of the experiment to exclude any underlying infection and to be allowed to acclimatize.

The animals were housed in cages and were maintained under controlled conditions of temperature ($24 \pm 2\,°C$) and light (12:12 h light:dark cycle). They received a standard basal diet (fat 5%, carbohydrates 65%, protein 20.3%, fiber 5%, salt mixture 3.7%, vitamins mixture 1%) and water ad libitum. The procedures of this experiment are compatible with the guide for care and use of laboratory animal approved by IACUC of Menoufia University, Egypt, Approval No:MNSP155.

Tartrazine (CAS 1934-21-0, Purity 86.7%) was purchased from Sigma Aldrich (Germany).

### Experimental design

The animals were randomly divided into two groups of 10 rats each.

Group 1: Animals were orally given distilled water 1 ml/kg b.wt. for 30 days and served as controls.

Group 2: Animals were orally given tartrazine 7.5 mg/kg b.wt.(dissolved in 1 ml of distilled water) daily for 30 days (*Himri et al., 2011*).

After 30 days of treatment, animals from control and treated groups were fasted overnight, and sacrificed under anesthesia. Blood samples were drawn from dorsal aorta into dry glass centrifuge tubes and left to clot, then centrifuged at 3,500 rpm for 15 min in a Beckman Model T-6 refrigerated centrifuge. Serum was separated and used for biochemical analysis.

Other blood samples were taken to isolate leucocytes for comet assay.

For histological and ultrastructure studies, small pieces of liver and kidney were quickly removed and fixed in an appropriate fixative.

### Biochemical analysis

For biochemical analysis, serum was rapidly separated by centrifuging the clotted blood at 3,000 g for 10 min in a Beckman Model T-6 refrigerated centrifuge and collected into new clean and dry tubes. Sera were stored at $-20\,°C$ until assayed for the biochemical

parameters. Aspartate aminotransferase (AST) and Alanine aminotransferase (ALT) were measured colorimetrically according to *Steven (1996)*, while Alkaline phosphatase (ALP) activity was determined according to the method described by *Mathieu et al. (1977)*.

Creatinine, urea and uric acid were estimated using the methods of *Henry (1974)*, *Patton & Crouch (1977)* and *Caraway (1955)*, respectively.

Malondialdehyde (MDA), Nitric oxide (NO) and total antioxidant capacity (TAC) were estimated by the methods of *Ohkawa, Ohishi & Yagi (1979)*, *Cortas & Wakid (1990)* and *Koracevic et al. (2001)*, respectively. They were determined using Bio-diagnostic assay kits according to the manufacturer's instructions (Giza, Egypt).

### Histological and ultrastructural study

After fixation of liver and kidney samples in Bouin's solution, they were dehydrated, cleared, embedded in paraffin wax and then prepared for histological examination using hematoxylin and eosin stain (*Bancroft & Gamble, 2002*).

For ultrastructural examination, small pieces of liver and kidney were immediately fixed in 4F1G in a phosphate buffer (pH 7.2) for 3 h at 4 °C, then post-fixed in 2% $OsO_4$ in the same buffer at 4 °C for 1–2 h. The specimens were dehydrated through a graded series of ethanol, embedded in epon-araldite mixture and polymerized at 60 °C. Ultrathin sections (50 nm) from selected areas were cut with glass knives on an LKB ultra microtome, double stained with uranyl acetate and lead citrate and examined using a JEOL 100CX electron microscope.

### Genotoxicity study
#### *Isolation of leucocytes for comet assay*

In order to isolate leucocytes, 4 ml of fresh blood was collected in tubes containing ethylenediaminetetraacetic acid (EDTA) as an anticoagulant. Blood samples were centrifuged at 10,000 rpm for 5 min. The white layer of leukocytes (buffy coat) was transferred to a new 2.0 ml microcentrifuge tube containing 2.0 ml of cold freezing mixtures (RPMI 1640 with 10% DMSO). Aliquots of 250 µl in 1.5 ml microcentrifuge tubes were stored in a −80 °C freezer until comet assay could be completed.

#### *Comet assay*

Alkaline Comet assay was performed according to *Collins (2004)*, with some modifications. The comets were visualized by a fluorescence microscope (Zeiss Axioplan 2) equipped with an Olympus C5050 camera. About 30 comets were scored and three different samples for each group were examined. Percentage of DNA in the tail was determined by the Open Comet software (*Gyori et al., 2014*).

## STATISTICAL ANALYSIS

The data of biochemical analysis and comet assay was expressed as mean ± SD of 3–5 replicates and were analyzed by one way ANOVA followed by student's $t$-test. The results are expressed as mean ± SD for three independent replicates. The difference was considered statistically significant when $p < 0.05$.
**Table 1** **Effect of tartrazine on serum biochemistry and oxidative biomarkers in male Wistar rats.**

| Group parameter | Control group | Tartrazine-treated group | $p$ value |
|---|---|---|---|
| Aspartate transaminase (U/ml) | $48.40 \pm 4.26$ | $128.40 \pm 18.97$ | 0.003* |
| Alanine transaminase (U/ml) | $29.40 \pm 4.07$ | $71.20 \pm 7.75$ | 0.001* |
| Alkaline phosphatase (U/l) | $47.0 \pm 5.18$ | $89.40 \pm 6.66$ | 0.001* |
| Creatinine (mg/dl) | $0.36 \pm 0.07$ | $0.72 \pm 0.06$ | 0.004* |
| Urea (mg/dl) | $38.20 \pm 1.77$ | $46.0 \pm 2.12$ | 0.02* |
| Uric acid (mg/dl) | $0.59 \pm 0.07$ | $0.92 \pm 0.07$ | 0.009* |
| Malondialdehyde(nM/ml) | $17.33 \pm 2.03$ | $46.67 \pm 5.61$ | 0.008* |
| Nitric oxide ($\mu$M/l) | $16.67 \pm 2.85$ | $41.0 \pm 3.79$ | 0.007* |
| Total antioxidants (mM/l) | $66.67 \pm 4.63$ | $28.33 \pm 4.06$ | 0.003* |

**Notes.**
Data represented as mean $\pm$ SD.
$p$: $p$ value for $F$ test (ANOVA) and significant between control and treated groups using Post Hoc Test (LSD).

## RESULTS

### Biochemical results

Table 1 shows that treatment with tartrazine resulted in a significant ($p < 0.05$) increase in the activity of plasma aspartate transaminase (AST), alanine transaminase (ALT) and alkaline phosphatase (AlP) compared to control. Also, our results showed a significant ($p < 0.05$) increase in plasma uric acid, urea and creatinine levels in tartrazine-treated animals. Additionally, rats exhibited a significant ($p < 0.05$) increase in plasma lipid peroxidation and nitric oxide (NO), and a significant decrease in the total of antioxidants after treatment with tartrazine.

### Light and electron microscopic results

Light microscopic observations showed distortion of hepatic architecture in liver sections of tartrazine-treated rats as compared to controls (Figs. 1A–1C). Most of hepatocytes appeared with necrotic nuclei and cytoplasmic vacuolization. Some cells had irregular-shaped nuclei, while others were devoid of nuclei (Fig. 1C). Moreover, the blood sinusoids revealed dilatation and congestion as well as white blood cell infiltration and Kupffer cells activation (Figs. 1B and 1C).

Electron microscope investigation revealed significant structural aberations in liver and kidney of tartrazine-treated rats. Nuclei of hepatocytes appeared irregular or pyknotic and others were with less electron dense chromatin as compared to control (Figs. 1D and 1E). Alterations in the lipid contents of cells were observed by the presence of numerous lipid droplets that vary in size and shape (Fig. 1E). Abnormal shaped mitochondria were observed with condensed opaque matrices and lack internal organization. The cisternae of rough endoplasmic reticulum appeared dilated and fragmented (Fig. 1F). Rarefied areas in the cytoplasm could be resulting from dissociation of cellular organelles as well as scattered ribosomes that were also observed (Fig. 1G).

Moreover, obvious histological changes were observed in the structure of the kidney in the animal group treated with tartrazine when compared to controls (Figs. 2A–2C). These changes includes degeneration in glomerular structure, loss of renal tubules integrity and

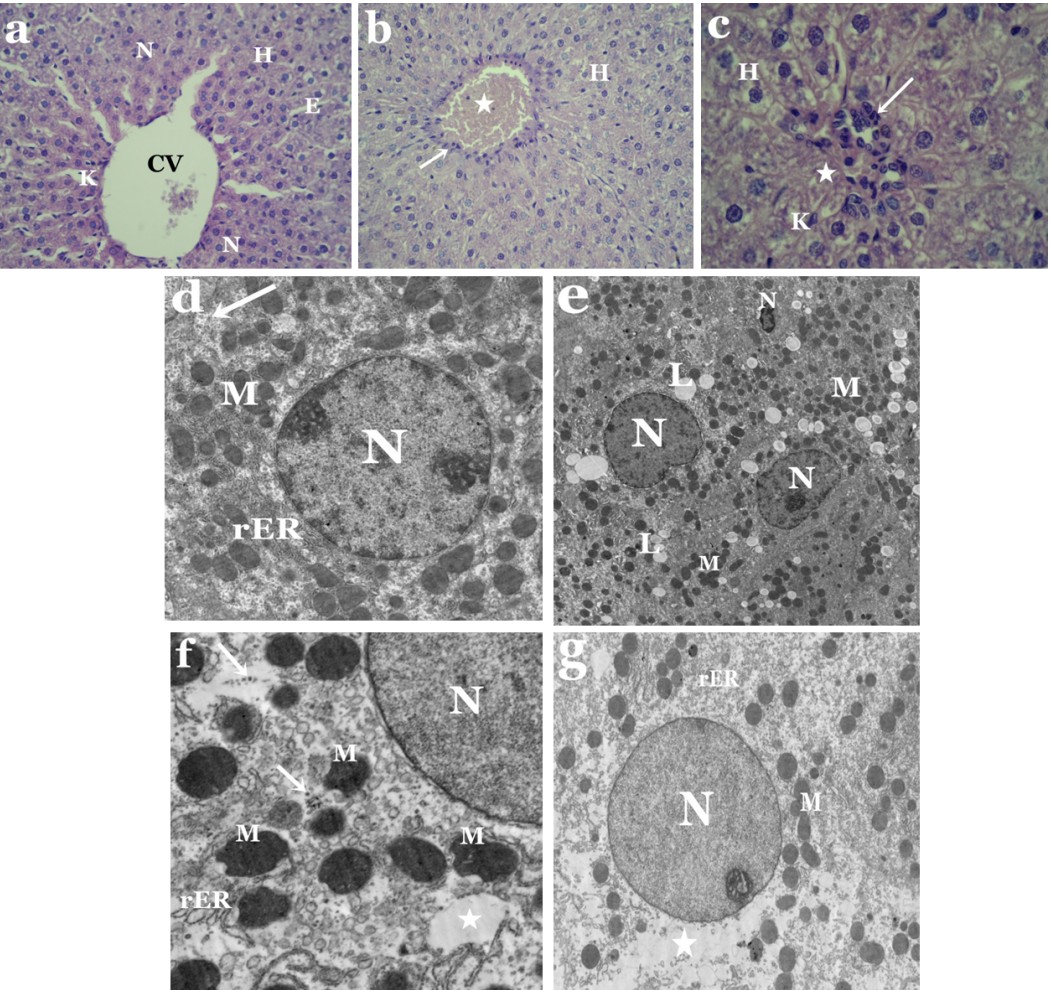

**Figure 1** (A–C) Light micrographs of liver sections from control and tartrazine-treated rats stained with H&E. (A) Section from control rat showing hepatocytes (H) with central spherical nucleus (N), central vein (CV), endothelial cells (E) and Kupffer cells (K). X400. (B) Section from rat treated with tartrazine showing necrosis of most hepatocytes (H), congestion of blood vessel (star) and leucocytic infiltration (arrow). X400. (C) Section from rat treated with tartrazine showing vacuolated hepatocytes (H), others devoid of nuclei (star), leucocytic infiltration (arrow) and Kupffer cells (K). X1000. (D–G) Electron micrographs of liver sections from control and tartrazine-treated rats. (D) Section from control rat, showing part of the hepatocyte with nucleus (N), rough endoplasmic reticulum (rER), numerous mitochondria (M) and glycogen particles (arrows). X2500. (E) Section from rat treated with tartrazine, showing irregular and pyknotic nuclei (N), numerous lipid droplets (L) and electron dense mitochondria (M). X1500. (F) Section from rat treated with tartrazine, showing abnormal shaped mitochondria (M), dilated rough endoplasmic reticulum cisternae (rER), scattered ribosomes (arrows) and clear area of cytoplasm (star). X4000. (G) Section from rat treated with tartrazine, showing less electron dense nucleus (N), clear area of cytoplasm (star), electron dense mitochondria (M) and fragmented endoplasmic reticulum cisternae (rER). X2000.
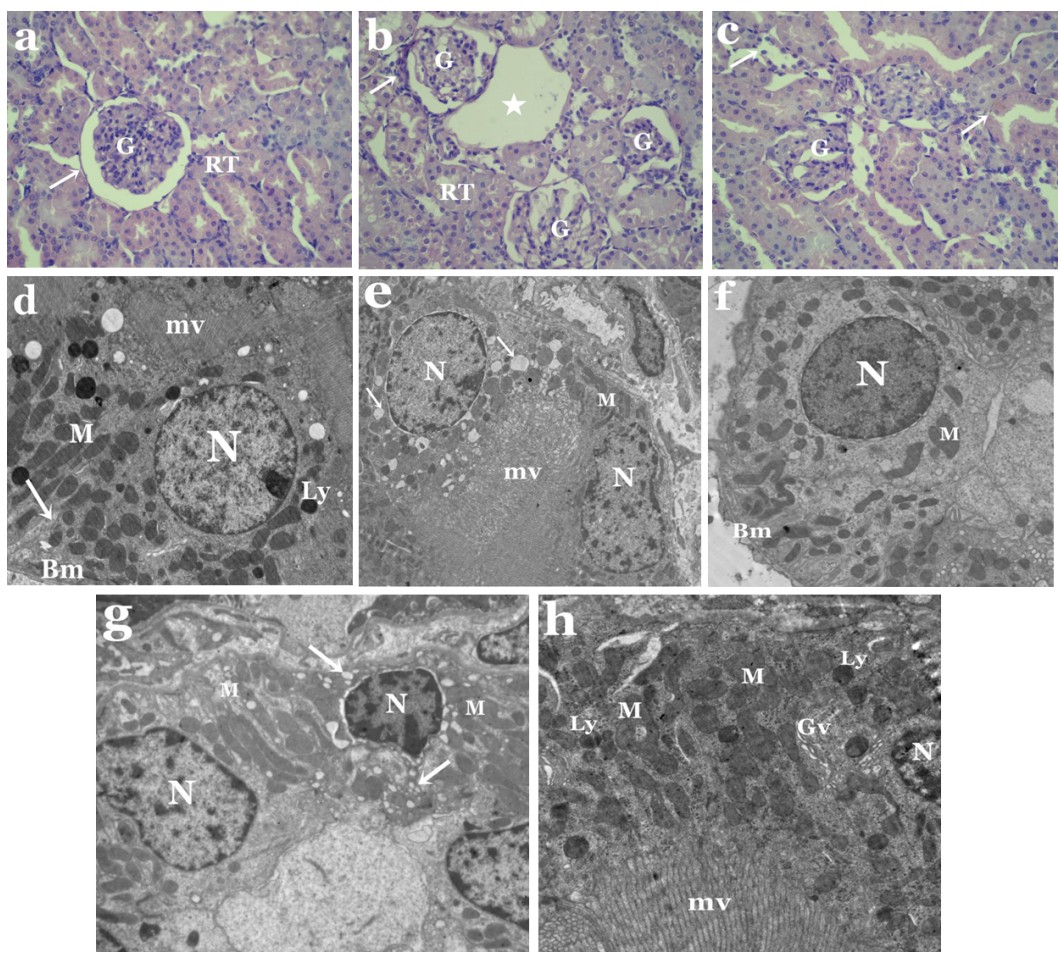

**Figure 2** (A–C) Light micrographs of kidney sections from control and tartrazine-treated rats stained with H&E. (A) Section from control rat showing Malpighian corpuscles with its glomerulus (G), Bowman's capsule (arrow) and renal tubules (RT). X400. (B) Kidney section from rat treated with tartrazine showing degenerated glomeruli (G), loss of renal tubules integrity (RT), huge cavity with fragmented areas (star) and inflammation (arrow). X400. (C) Kidney section from rat treated with tartrazine showing damage in renal tubules membrane (arrows) and degenerated glomeruli (G). X400. (D–H) Electron micrographs of kidney sections from control and tartrazine-treated rats. (D) Kidney section from control rat showing proximal tubular cells with apical microvilli (mv), basement membrane (Bm), basal infoldings (arrow), nucleus (N), numerous mitochondria (M) and lysosomes (Ly). X2000. (E) Kidney section from rat treated with tartrazine showing disrupted proximal tubular cells with irregular nucleus (N), vacuolated cytoplasm (arrows) and disordered mitochondria (M). X2000. (F) Kidney section from control rat showing distal tubular cell with nucleus (N), mitochondria (M) and basement membrane (Bm). X2000. (G) Kidney section from rat treated with tartrazine showing disrupted distal tubular cells with nucleus (N), mitochondria (M) and vacuoles (arrows). X 2500. (H) Enlarged part of proximal tubular cell from rat treated with tartrazine showing part of pyknotic nucleus (N), destructed mitochondria (M), numerous lysosomes (Ly), Gologi vesicles (Gv) and microvilli (mv). X4000.

the presence of areas of huge vacuoles (Fig. 2B). Membrane injury in apical surfaces of tubular epithelial cells and degeneration in the basal membrane of cells were also noticed (Fig. 2C).

Electron microscope investigation showed remarkable ultrastructural alterations in the kidney of rats treated with tartrazine. These changes includes degenerated proximal and distal tubular cells with nuclei containing clumps of marginated heterochromatin (Figs. 2D and 2F). Some of these nuclei appeared with irregular outline,while others were pyknotic. Vacuolated cytoplasm was a prominent feature in both proximal and distal tubular cells (Figs. 2E and 2G). Also, pleomorphic and disorganized mitochondria as well as increased number of lysosomes and Golgi vesicles were observed in proximal tubular cells (Fig. 2H). In distal tubular cells, some mitochondria appeared normal while others were disrupted (Fig. 2G).

### Genotoxicity results

Comet assay resulted that tartrazine possesses a genotoxic effect in the white blood cells of treated rats (Figs. 3A and 3B). This genotoxic effect was observed as a significant increase ($p < 0.05$) in the percentage of DNA in the comet tail in the nuclei of leucocytes of tartrazine-treated animals as compared to controls.

## DISCUSSION

In the present work, administration of tartrazine significantly elevated the activity of serum ALT, AST and ALP. The increased levels of liver enzyme activities in blood is an indication of possible tissue damage. These results are in accordance with data reported by other investigators who attributed similar changes in liver function to hepatocellular impairment. This liver damage would releases greater than normal levels of intracellular enzymes into the blood (*Amin, Abdel Hameid & AbdElsttar, 2010*; *Himri et al., 2011*; *El-Wahab & Moram, 2015*; *Saxena & Sharma, 2015*).

Serum levels of creatinine and urea are major factors in determination of both, the glomerular filtration efficacy and proximal tubular secretion rate (*Azu et al., 2010*). Such damage in the the filtering compartments of kidney, blood levels of creatinine and urea increase. Our results show a significantly elevated serum levels of creatinine, urea and uric acid in tartrazine-treated group as compared to controls. These results are in agreement with *Tawfek et al. (2015)* who found a significant increase in serum creatinine and urea in rats following the consumption of different types of food additives including tartrazine, sunset yellow and sodium benzoate. Similar finding was observed by *Ashour & Abdelaziz (2009)* in rats dosed with organic azo dye (fast green) orally for 35 days. In addition, these results are in agreement with data reported by *Amin, Abdel Hameid & AbdElsttar (2010)* who revealed a significant increase in serum creatinine and urea levels in rats after consumption of tartrazine. Recently, *Nabila et al. (2013)* mentioned that tartrazine application induced significant elevations in urea and creatinine levels that is associated with impaired renal function and the inability of the kidney to filter body fluids.

Oxidative stress is referred to a reactive oxygen species (ROS)/antioxidant imbalance. It occurs when the overall level of ROS exceeds the potential of the antioxidants. Thus,

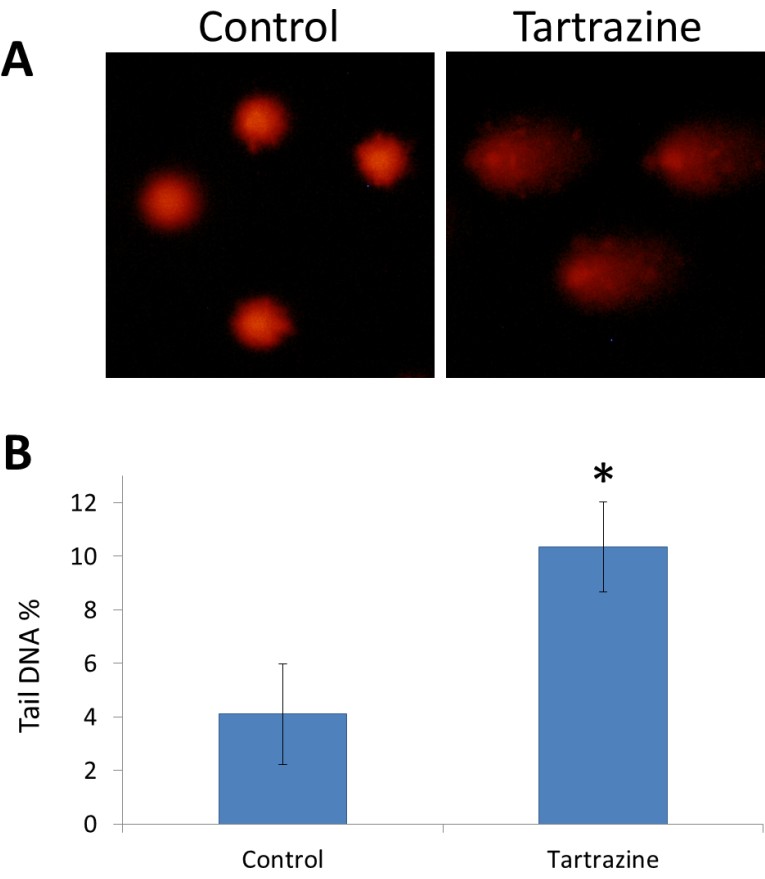

**Figure 3** **The genotoxic effect of tartrazine on leucocytes of rats.** (A) Fluorescence micrograph representing nuclei of leucocytes after the comet assay. Control nuclei of untreated animals appear intact with no detectable DNA damage. Nuclei of tartrazine-treated animals appear damaged. (B) Bar graph showing the tail DNA damage in percentage in nuclei of leucocytes of control and tartrazine-treated animals subjected to comet assay. Data are Mean $\pm$ SD ($n = 3$, $t$-test, $^*p < 0.05$).

oxidative stress may occur because of accelerated ROS production, a drop of the antioxidant mechanisms, or both (*France's et al., 2013*). In the present study, elevated levels of malondialdehyde (MDA, end product of lipid peroxidation) and nitric oxide (NO) clearly indicates oxidative stress occurrence in the tartrazine-treated rats. These results are in hand with data reported by *Omca, Zhang & Ercal (2012)*, who studied the oxidative effects of tartrazine and other azo dyes on Chinese hamsters. The increased lipid peroxidation may be attributed to the generation of ROS that result from tartrazine administration. Because tartrazine belongs to the group of azo dye food colorants,it is metabolized inside the body into aromatic amines by intestinal microflora. These formed amines are able to generate ROS as part of their metabolism by the interaction of the active amino groups with nitrite or nitrate containing foods (*Moutinho, Bertges & Assis, 2007*). NO is considered as another important source of free radicals that might contribute to alterations in energy metabolism. *Peresleni et al. (1996)* demonstrated that oxidative stress to epithelial cells increases NO syntheses which results in elevated NO release, nitrite production and decreased cell viability.
The increased total tissue antioxidant capacity (TAC) provides a gross estimation of how the body can react against oxidative stress (*Costantini & Verhulst, 2009*).

In the current work, the decrease in the level of TAC in the serum of tartrazine-treated animals may be related to increased free radical generation due to tartrazine administration and/or impaired antioxidant machinery leading to increased oxidative stress.

Histological and ultrastructural results clearly showed that administration of tartrazine had led to distortion of hepatic architecture as well as degeneration in kidney structure of rats. Light microscopic figures showed tartrazine-induced necrosis of most hepatocytes, congestion of blood sinusoids, infiltration of white blood cell, activated Kupffer cells, damaged glomerular and renal tubule membranes. These findings are in agreement with *Himri et al. (2011)*, *Mehedi et al. (2013)* and *Saxena & Sharma (2015)* who indicated that tartrazine administration alters the histological structure of livers and kidneys in experimental animals. *Rus et al. (2010)* found that tartrazine and carmoisine causes congestion, stasis and edema in liver and kidney of Guinea pigs leading to apoptosis in hepatocytes and atrophy of renal structures.

Several ultrastructural alterations were recorded in the livers and kidneys of rats treated with tartrazine. The hepatocytes and renal tubule epithelium appeared with irregular or pyknotic nuclei. Large numerous lipid droplets, disorganization of mitochondria, rough endoplasmic reticuli (rER) and degenerated cytoplasmic areas were observed in the cytoplasm of hepatocytes. Meanwhile, proximal and distal tubular cells possess vacuolated cytoplasm and defective mitochondria. These results are similar to those of previous studies conducted with food additives, sodium benzoate and citric acid (*Bakar & Aktac, 2014*; *Chen et al., 2014*; *Aktac et al., 2008*; *Kaboglu & Aktac, 2002*). The presence of pyknotic nuclei and necrosis of hepatocytes and tubular cells in the current work clearly indicates toxicosis as previously described by *Deveci et al. (2011)* and *Sarkar & Ghosh (2012)*. Structural changes in nuclei and ER, the critical structures for biosynthesis of glycoconjugates, could probably impairs glycosylation mechanisms and hence affect cell surface sialylation (*Bakar & Aktac, 2014*).

The increased amount of hepatocellular lipids was previously reported by *Sinha & D'Souza (2010)* and *Khidr, Makhlouf & Ahmed (2012)* after sodium benzoate administration in experimental animals. *Cheville (1988)* reported that toxins may affect ribosomes and their ability to produce peptide chains and decreases the amount of proteins involved in the transport of triglycerides, that are produced at their normal rates, causing accumulation of lipid globules. Deformation of mitochondria and rER in hepatocytes and renal tubular cells cytoplasm after tartrazine administration could be a response to chemical stressors (*Khidr, Makhlouf & Ahmed, 2012*). *Reyes, Valim & Vercesi (1996)* found that synthetic food coloring agents such as tartrazine, inhibited mitochondrial respiration in liver and kidney of rats. It also affected the integrity of mitochondrial membranes, which is critical for maintaining vital mitochondrial functions and determination of apoptosis in cells. *Bakar & Aktac (2014)* reported that degeneration of mitochondrial membranes and cristae negatively affect oxidative metabolism of cells.

Our electron microscopic results showed a clear destruction of hepatocytic cytoplasm. *Sinha & D'Souza (2010)* reported that damage from toxic insults may cause swallowing of hepatocytes with large clear spaces. The integrity of cytoplasm is important for regular

 

intracellular trafficking which in turn would be damaged due to cytoplasmic vacuolization (*Bakar & Aktac, 2014*). The distortion of renal structures and tubular vacuolization found in the present work was previously described by other workers in kidneys of animals treated with the food additive, monosodium glutamate (*Eweka, 2007*; *Abass & Abd El-Haleem, 2011*; *Afeefy, Mahmoud & Arafa, 2012*).

Concerning the genotoxicity of synthetic colors, the obtained results revealed that tartrazine caused DNA damage in leucocytes as detected by comet assay. This genotoxic effect is probably due to the direct contact of tartrazine with nuclear DNA (*Himri et al., 2012*). Data pertaining to the genotoxic effect of tartrazine with positive results are available. This finding agrees with *Mpountoukas et al. (2010)* who investigated the toxic effect of tartrazine at 0.02–8 mM in human peripheral blood cells *in vitro*. In addition, tartrazine has been shown to induce chromosomal aberrations in fibroblast cells of *Muntiacus muntjac* (*Chung, Fulk & Andrews, 1981*). *Hassan (2010)* also revealed that administration of a daily dose of tartrazine (7.5 and 15 mg/kg b.wt.) for seven weeks leads to liver and kidney DNA damage.

On one hand, a study carried out by *Poul et al. (2009)* showed that acute oral administration of tartrazine did not induce genotoxic alterations in the micronucleus gut assay in mice at doses up to 2 g/kg b.wt. On the other hand, induction of tartrazine-induced DNA damage in comet assays was observed in cells from the colon of mice (*Sasaki et al., 2002*) at a dose that is slightly higher than the recommended human daily intake approved by the Joint FAO/WHO Expert Committee on Food Additives (*MHW, 1999*). Therefore, further comet assay investigations performed according to the latest recommended protocol (*Hartmann et al., 2003*) might be useful to clarify these discrepancies.

According to the above-discussed results, it can be concluded that tartrazine is able to generate ROS thus accelerating oxidative stress, altering the structure and biochemical profiles in hepatic and renal tissues. Therefore, controlling the consumption of tartrazine is important for the health and limiting the use of tartrazine, especially in foods used by children, is highly advisable.

### Funding
This work was supported by the Institute of Scientific Research and Revival of Islamic Heritage at Umm-Al Qura University, project number 43405033. The funders had no role in study design, data collection and analysis, decision to publish, or preparation of the manuscript.

### Grant Disclosures
The following grant information was disclosed by the authors:
Institute of Scientific Research and Revival of Islamic Heritage at Umm-Al Qura University: 43405033.

### Competing Interests
The authors declare there are no competing interests.
## Author Contributions

- Latifa Khayyat analyzed the data, wrote the paper, reviewed drafts of the paper.
- Amina Essawy conceived and designed the experiments, performed the experiments, contributed reagents/materials/analysis tools, wrote the paper, reviewed drafts of the paper.
- Jehan Sorour performed the experiments, contributed reagents/materials/analysis tools, wrote the paper.
- Ahmed Soffar performed the experiments, analyzed the data, prepared figures and/or tables.

## Animal Ethics

The following information was supplied relating to ethical approvals (i.e., approving body and any reference numbers):

The procedures of this experiment are compatible with the guide for care and use of laboratory animal approved by IACUC of Menoufia University, Egypt, Approval No:MNSP155.

## Data Availability

The raw data has been supplied as a Supplemental File.

## Supplemental Information

Supplemental information for this article can be found online at http://dx.doi.org/10.7717/peerj.3041#supplemental-information.

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
