# Peer review of "Tartrazine induces structural and functional aberrations and genotoxic effects in vivo"

_PeerJ, doi:10.7717/peerj.3041_

## Round 0.1 · original submission · Minor Revisions

Our reviewers are generally pleased with the scientific content of your manuscript. Please address the remaining concerns, specifically:
from reviewer #2

"There are many previous reports pertaining the toxicity of tartrazine in animal system then what is the novelty of the present work?
In your study animals were orally given tartrazine 7.5mg/kg b.wt. (dissolved in 1 ml of distilled water) daily for 30 days. Is all animals alive till the 30th day? is any other health problems observed in the mice during this period when compared to the control group? Out of the 10 mice studied how much are subjected to further biochemical, histological and genotoxic studies?"

from reviewer #3:

-discuss/compare your results with those of Saxena and Sharma (DOI: 10.4103/0971-6580.172286)
"Why did you use only one dose of tartrazine, and very low. In literature higher doses of tartrazine were used. "


The manuscript contains many typographical errors, several of which have helpfully been highlighted by the reviewers, and which should also be corrected before acceptance.


Review-level comments by the Editor:

- Please state the rationale behind the choice of your biochemical markers. Were any addition biochemical markers measured and found to be unchanged? If so, please include them in Table 1.

- I think table 1 would be more readable if no abbreviations were used.

- Please state the frequency of abnormal histological features in each group of animals. Further, was the researcher who performed the histological observations "blinded" with regard to the origin of the samples? if so, please state that. If not, please perform blinded comparisons of morphological features (with data from several animals from each group).

in the discussion, please state the tartrazine concentrations used in the other studies. As mentioned by reviewer #3, you used very low concentrations but still observed toxic effects, whereas some studies did not observe toxicity with much higher doses. You should clearly state these discrepancies, and possibly suggest ways to explain/reconcile these results.

line 292: the reference for " In addition, tartrazine has been shown to induce chromosomal aberrations in fibroblast cells of Muntiacus muntjac(Mpountoukas et al., 2010)" is wrong. It should be: Chung, K.T., Fulk, E.G., Andrews, A.W., 1981. Mutagenicity testing of some commonly used dyes. Appl. Environ. Microbiol. 42, 641–648

Reviewer 1 ·

Basic reporting

Clear and unambiguous

Experimental design

Experimental design is well thought out.

Validity of the findings

Findings are valid.

Additional comments

The research is well planned and the results can be published.

·

Basic reporting

At first, I would like to thank you for choosing me as a referee of your distinguished journal, concerning the manuscript entitled ‘Tartrazine induces structural and functional aberrations and genotoxic effects in-vivo.
This paper investigated the toxic potential of a synthetic food colorant Tartrazine in animal test system. The topic of research is interesting and nicely designed, however there are several typographical, spacing and language problems throughout the manuscript. Some of the sentences are meaningless which need rephrasing. Hence this manuscript is not acceptable in its present form, needs major revision

Experimental design

The experimental design was found to be good

Validity of the findings

The tables and figures were checked with raw data and found to be valid

Additional comments

• Authors should be updated the references because a lot of reference was belongs to previous years.
• Authors should add CAS Number of the tartrazine in the methodology section
• There are many previous reports pertaining the toxicity of tartrazine in animal system then what is the novelty of the present work?
• In your study animals were orally given tartrazine 7.5mg/kg b.wt. (dissolved in 1 ml of distilled water) daily for 30 days. Is all animals alive till the 30th day? is any other health problems observed in the mice during this period when compared to the control group? Out of the 10 mice studied how much are subjected to further biochemical, histological and genotoxic studies?
• Spacing problem in heading –‘Tartrazineinduces’ give space between the words ‘tartrazine’ and ‘induces’
• Heading: Check the hyphen between in-vivo is it needed
• Line spacing problem throughout the manuscript, correct it and make it uniform as per the journals guidance
• Abstract ‘adverseeffect’ give space between
• Abstrcat ‘decreasedlevel’ give space between
• Abstract: restructure the sentence ‘Our results showed………. compared to controls
• Abstract: tartrazinewas, give space between
• Abstract: withseverehistopathological, give space betweenk ; check the word severe/several
• Abstrcat: change the sentence as - Taken together, the results showed that tartrazine intake may leads to an adverse health effects.
• Keywords: Change genotoxic with genotoxicity
• Use either color or colour, keep uniformity in language used
• Introduction: line 1 colour , remove space between colour & ,
• Introduction: line 2 colorantfor add space between
• Rewrite the sentence, A partial list……. and chewing gums.
• Introduction line 4 remove ‘also’
• (Amin et al.,2010) add space after ,
• Intro line 7 ‘alow’ add space between
• Intro 1st para last line cooking(Mehediet al., 2009). Give space between cooking and refer cited, and also between Mehedi and et
• Intro second para 1st and 6th para spacing problem
• Restructure the 3r paragraph
• Intro final para several spacing problems please rectify, change ‘effect of this food coloring on’ as effect of this food colorant on
• Materi & meth : Line 3 spacing problem
• Materi & meth: They received a standard… change to Animals were fed with a standard…
• Biochemical analysis: ‘processed for determination into clean and dry tubes’ sentence not clear rephrase it
• Check the citation pattern of the journal and make it uniform
• Please check the indentation of paragraph in journal’s instruction, if needed provide
• Biochemi analy: last para Cortas&Wakid add space between
• Isolation of leucocytes for comet assay: ‘Aliquots of 250 µl in 1.5 ml microcentrifuge tubes were frozen by placing in a -80°C freezer until comet assay could be completed’ sentence not clear
• Statistical analysis: spacing problem ‘student’s t-test.Statistical’ and ‘mean± SD’
• Results: ‘plasma uric acid urea’ add comma
• Light and electron microscopic results: correct spacing problems throughout paragraphs, tocontrols; nuclei(Fig.1c).Moreover; abbarationsin;
• There is several spacing problems are there between lines and words please read carefully and rectify all. It is found difficult to mention each and every spacing problem
• Light and electron microscopic results:3rd paragraph remove ‘Moreover, obvious’
• Spacing problems in figure legends
• Fig. (2): (a-c) LM & EM abbreviations should be mentioned in the first use, in figure 1 mentioned as ‘Light micrographs’ keep uniformity throughout the manuscript
• Fig. (2): legend Malpighian, M should be lower case letter
• Genotoxicity results ‘Using the comet assay, the present results show that…’ Sentence should be change as ‘Comet assay resulted that tartrazine…….
• Sentence too long, change as ‘This genotoxic effect was observed as a significant increase (p< 0.05) in the percentage of DNA in the comet tail of the nuclei of leucocytes compared to controls.
• Check the journal format to write ‘ figure’ in the legend and citation in the text, in the first two figure legends, it was used as Fig.(1): & Fig. (2): but in the 3rd figure legend it was used as Figure 3. Keep uniformity
• Figure 3 legend, change the sentence as ‘(B) Graph showing the tail DNA damage percentage in leucocyte nuclei of control and tartrazine-treated animals after comet assay.
• Discussion line 1 spelling correction ‘admiistraiton’
• Rectify the spacing problem throughout discussion
• Discussion 2nd para, 3rd line change the sentence as ‘Damages in the filtering compartments of kidney results elevated levels of creatinine and urea in the blood.
• Discussion 2nd para, change the sentence as ‘The results of the present study showed that elevated levels of creatinine, urea and uric acid in the serum of tartrazine-treated group when compared to control.
• Discussion 2nd para line 8 remove bracket, ‘benzoate)’
• Discussion 3rd para remove the sentence ‘,adropof the antioxidant mechanisms, or both’
• Discussion 2nd para line 11, chage as ‘intestinal micro flora’
• Discussion para 5 sentence should be changed as ‘Light microscopic figures showed tartrazine-induced necrosis of most hepatocytes, congestion of blood sinusoids, infiltration of white blood cell, activated Kupffer cells, damaged glomerular and renal tubule membranes.
• Discussion para 6 remove comma ‘reticuli (rER),’
• Discussion para 6 sentence changed as ‘The presence of pyknotic nuclei, necrosis of hepatocytes and tubular cells in the current work clearly indicates toxicosis as previously described by Deveci et al. (2011) andSarkar and Ghosh (2012).
• Discussion para 6, sentence change as ‘Reyes et al. (1996) found that synthetic food coloring agents such as tartrazine, inhibited mitochondrial respiration in liver and kidney of rats. It also affected the integrity of mitochondrial membranes, which is critical for maintaining vital mitochondrial functions and determination of apoptosis in cells.
• Change as ‘Our electron microscopic results showed a clear’
• Sentence should be changed as ‘Hassan et al. (2010) also revealed that adminatration of a daily dose of tartrazine (7.5 and 15 mg/kg b.wt.) for 7 weeks leads to liver and kidney DNA damage.
• References, correct the spacing problems throughout, and are not in a uniform format. Check the journals format for references and correct it.

Reviewer 3 ·

Basic reporting

I regret to inform you that the manuscript in its present form has been found to have a number of spelling and typographical errors that make comprehension difficult since there is not a space between two words in most cases. The manuscript needs to be carefully checked before publication.

Some recent references are missing in the same fild eg. Saxena B, Sharma S. (2015) Food Color Induced Hepatotoxicity in Swiss Albino Rats, Rattus norvegicus. Toxicol Int. 22(1):152-7. doi: 10.4103/0971-6580.172286.

No comment

No comment

Experimental design

No comment

In the experimental design it will better to include a second dose of tartrazine used.


No comment


No comment

Validity of the findings

The findings are very interesting. Comparison of these results with other (and this is important) showed that the dose of tartrazine administered in rats was very low and even exert its toxic effect in biochemical markers (Saxena B, Sharma S. (2015) Food Color Induced Hepatotoxicity in Swiss Albino Rats, Rattus norvegicus. Toxicol Int. 22(1):152-7. doi: 10.4103/0971-6580.172286.)

No comment

No comment


No comment

Additional comments

The results are very interesting.

Annotated reviews are not available for download in order to protect the identity of reviewers who chose to remain anonymous.

---

## Round 0.2 · accepted · Accept

No further changes to the scientific content of the manuscript seem needed. The issues noted by Reviewer #2 should be resolved in production.

·

Basic reporting

This paper investigated the toxic potential of a synthetic food colorant Tartrazine in animal test system. The topic of research is interesting and nicely designed, however there are still several spacing problems throughout the manuscript including the references. Hence this manuscript is not acceptable in its present form, needs minor revision

Experimental design

The experimental design was found to be good

Validity of the findings

The tables and figures were checked with raw data and found to be valid

Additional comments

There are still spacing problem throughout, it will not suitable for your journal in the present format. In the reference list also many spacing problems and most one are not in the journal format. so it need through revision carefully.

Reviewer 3 ·

Basic reporting

No comments

Experimental design

No comments

Validity of the findings

No comments

Additional comments

The article should be accepted as is